# Mechanisms of Variation in Abdominal Adipose Color Among Male Kazakh Horses Through Non-Coding RNA Sequencing

**DOI:** 10.3390/biology14091285

**Published:** 2025-09-17

**Authors:** Yuhe Zhou, Xinkui Yao, Jun Meng, Jianwen Wang, Yaqi Zeng, Linling Li, Wanlu Ren

**Affiliations:** 1College of Animal Science, Xinjiang Agricultural University, Urumqi 830052, China; 15160990694@163.com (Y.Z.); yaoxinkui@xjau.edu.cn (X.Y.); junm86@sina.com (J.M.); dkwjw@xjau.edu.cn (J.W.); xjauzengyaqi@163.com (Y.Z.); lilinling@xjau.edu.cn (L.L.); 2Xinjiang Key Laboratory of Equine Breeding and Exercise Physiology, Urumqi 830052, China

**Keywords:** Kazakh horse, abdominal adipose tissue, LncRNA, miRNA, circRNA, ceRNA network

## Abstract

The Kazakh horse is an outstanding indigenous breed and a valuable genetic resource, characterized by diverse applications and a superior meat quality. However, the reason as to why the abdominal adipose tissue appears either white or yellow remains unclear. Consumer surveys have shown a general preference for yellow fat. To investigate the mechanisms underlying this color variation, we sequenced and analyzed non-coding RNAs from the abdominal adipose tissue of 16 Kazakh horses. The results identified multiple differentially expressed non-coding RNAs between yellow and white fat as well as key genes and signaling pathways associated with fat color, providing preliminary insights into the molecular basis of adipose color differences in Kazakh horses.

## 1. Introduction

The Kazakh horse is a significant indigenous breed in China, characterized by a large population and wide geographic distribution [1]. It possesses stable genetic traits, strong adaptability, and a high tolerance for coarse forage [2]. At present, this breed is extensively utilized for both dairy and meat production [3], serving as a major source of horse meat in China. In recent years, consumer demand for horse meat has been steadily increasing [4]. As a key component of meat, fat plays an important role in quality evaluation. Consumers generally associate yellow fat with pasture-based rearing rather than intensive fattening in feedlots, and therefore tend to prefer yellow fat. As a result, fat color has become an important criterion in assessing meat quality [5,6].

Adipose tissue is a type of loose connective tissue widely distributed throughout the animal body, composed of aggregated adipocytes. It plays an important role in regulating nutritional status [7] and mediating immune responses [8] and constitutes an essential component of the organism. Based on differences in color and biological function, adipose tissue is classified into white adipose tissue (WAT) and brown adipose tissue (BAT) [9]. WAT primarily serves as an energy reservoir by storing triglycerides, whereas BAT generates heat by metabolizing fatty acids and glucose when energy demand is high [10]. In addition, WAT can undergo browning to form beige adipose tissue [11].

Substantial progress has been made in adipose-related research across a variety of animal species including cattle [12], sheep [13], chickens [14], and pigs [15]. Through integrated transcriptomic and metabolomic analyses, Chen et al. [16] identified ten key genes, such as *PLA2G5*, *PLA2G2D*, and *PTGES*, that regulate lipid-associated signaling pathways involving linoleic acid, arachidonic acid, tyrosine, and retinol. These genes were found to reduce subcutaneous fat deposition and alter the fatty acid composition in Tibetan sheep. Peng et al. [17] demonstrated that genes such as *ACC1* and *FASN* acted synergistically to promote the synthesis of palmitic acid, which was subsequently desaturated by *ELOVL6* and *SCD* to generate long-chain fatty acids essential for triglyceride synthesis. Additionally, *DGAT2* enhances triglyceride synthesis, while *PLIN1* regulates triglyceride storage within lipid droplets, collectively contributing to the enhanced triglyceride synthesis and storage capacity in Laiwu pigs. Furthermore, increasing attention has been directed toward the role of ncRNAs in regulating adipose tissue. Zhang et al. [18] reported that the elevated expression of LncRNA *TCONS_00161198* and its target genes *CSF3R* and *ACC* in Kele pigs led to increased triglyceride and promoted adipose deposition. Li et al. [19] found that *ADNCR* was predominantly localized in the cytoplasm of adipocytes for Qinchuan cattle, where it competes with *miR-204* for binding, significantly regulating the expression of *SIRT1*. This interaction downregulates the expression of adipogenic genes such as *PPARG* and *FABP4*, thereby inhibiting adipocyte differentiation.

At present, the molecular regulatory mechanisms underlying the color variation in the adipose tissue of Kazakh horses remain unclear. This study utilized ncRNA sequencing to profile and analyze abdominal adipose tissue samples exhibiting different colors from male Kazakh horses, with the aim of investigating key genes and biological processes involved in regulating color variation. The results are expected to provide a foundational framework and theoretical basis for future research on adipose tissue color variation in equine species.

## 2. Materials and Methods

### 2.1. Sample Collection

In 2025, 16 adult male Kazakh horses aged 6 to 8 years were randomly selected from the Wantong Animal Husbandry Co. Ltd., Emin County. Abdominal adipose tissue samples were collected from these animals. According to the color, the samples were categorized into two groups (i.e., Group W (white adipose tissue) and Group Y (yellow adipose tissue)) with eight biological replicates in each group. All experimental horses were fed high-quality alfalfa hay and corn kernels, with ad libitum access to water. All horses were raised, managed, fattened, and slaughtered under uniform standardized conditions. Immediately post-slaughter, abdominal adipose tissue samples measuring 1 cm × 1 cm × 1 cm were excised from the left half of the carcass, placed into cryogenic vials with two aliquots per sample, rapidly frozen in liquid nitrogen, and stored at −80 °C for analyses.

### 2.2. Non-Coding RNA Sequencing

The total RNA was extracted from each adipose tissue sample using TRIzol reagent (Thermo Fisher Scientific, Carlsbad, CA, USA). RNA concentration and integrity were evaluated with a NanoDrop microvolume spectrophotometer (Thermo Fisher Scientific, Carlsbad, CA, USA), an Agilent 2100 Bioanalyzer (Agilent Technologies, Santa Clara, CA, USA), and 1% agarose gel electrophoresis. Ribosomal RNA (rRNA) was removed using the Ribo-off rRNA Depletion Kit (Vazyme, Nanjing, China), and sequencing libraries were prepared with the VAHTS Universal V6 RNA-Seq Library Prep Kit (Vazyme, Nanjing, China). The main steps were as follows: after the hybridization of RNA with probes, rRNA was digested by RNase H and DNase I, and the resulting rRNA-depleted RNA was purified with VAHTS RNA Clean Beads; the purified RNA was then fragmented and used for first-strand cDNA synthesis with random primers, followed by second-strand cDNA synthesis in a dUTP-containing system; Illumina adapters were ligated using Rapid DNA Ligase, and the ligated products were purified twice with VAHTS DNA Clean Beads; PCR amplification was carried out with a premix containing the UDG enzyme to enrich adapter-ligated fragments while degrading the second-strand cDNA; the amplified products were further purified with magnetic beads to generate high-quality libraries. Library quality was assessed using the High Sensitivity DNA Assay Kit (Agilent Technologies, Santa Clara, CA, USA), and precise quantification was performed via qPCR. Sequencing was conducted by Kidio Biotechnology Co. Ltd. (Guangzhou, China) [20].

### 2.3. Quality Control and Assessment of Data

Raw sequencing data in FASTQ format, containing adapter sequences and low-quality reads, were obtained directly from the sequencer. To obtain high-quality clean reads, filtering was performed using fastp (v 0.23.4) with the following parameters: (1) removal of reads containing adapter sequences; (2) removal of reads with >10% ambiguous bases (N); and (3) removal of low-quality reads in which bases with a Phred quality score ≤20 accounted for >50% of the sequence [21]. Differential expression analysis of lncRNA, miRNA, and circRNA transcripts was conducted, and significantly differentially expressed transcripts were identified using the criteria |log2(fold change)| ≥ 1.5 and *p* < 0.05, with multiple testing correction applied to *p*-values using q < 1.00.

### 2.4. Differential Expression Analysis

For each sample, aligned reads were assembled using a reference-based method with StringTie v2.2.1. For each transcript region, FPKM (fragments per kilobase of transcript per million mapped reads) values were calculated to quantify the gene expression levels and their variability. RNAs with *p* < 0.05 and an absolute fold change greater than 1 were considered significantly differentially expressed.

### 2.5. GO and KEGG Enrichment Analysis

Enrichment analysis was performed to functionally annotate differentially expressed ncRNAs (DEncRNAs) using GO and KEGG pathway analyses. KOBAS was employed to assess the enrichment of DEncRNAs in KEGG pathways, while GO functional enrichment analysis was conducted using GOseq. Terms with *p* < 0.05 were considered significantly enriched.

### 2.6. Construction of the ceRNA Network

A competing endogenous RNA (ceRNA) network comprising differentially expressed long non-coding RNAs (DELncRNAs), microRNAs (DEmiRNAs), and circular RNAs (DEcircRNAs) was constructed through the following procedures: (1) interactions between DELncRNAs and DEmiRNAs were predicted using miRanda v3.3a; (2) miRDB and TargetScan v8.0 were employed to predict interactions between DEmiRNAs and DEcircRNAs; and (3) overlapping DELncRNAs and DEmiRNAs were identified, and an integrated ceRNA network comprising interactions among lncRNAs, miRNAs, and circRNAs was generated using Cytoscape v3.9.

### 2.7. RT-qPCR Validation

To validate the selected LncRNAs, miRNAs, and circRNAs, the total RNA was reverse-transcribed into complementary DNA (cDNA). The procedure was conducted as follows. Pre-chill a grinding tube on ice and add 1 mL of RNA extraction reagent along with three 3 mm grinding beads. Add 5–20 mg of tissue and thoroughly homogenize until no visible tissue fragments remain. Centrifuge the homogenate at 12,000 rpm and 4 °C for 10 min and collect the supernatant. Add 100 μL of chloroform substitute, invert the tube for 15 s to mix, and incubate at room temperature for 3 min. Centrifuge again at 12,000 rpm and 4 °C for 10 min, transfer 400 μL of the upper aqueous phase to a new tube, and add 550 μL of isopropanol, gently inverting to mix. Incubate at −20 °C for 15 min. Centrifuge at 12,000 rpm and 4 °C for 10 min; a white pellet at the bottom represents RNA. Discard the supernatant and wash the pellet with 1 mL of 75% ethanol, centrifuge at 12,000 rpm and 4 °C for 5 min, and repeat the wash once. Completely remove the ethanol and air-dry the pellet in a sterile workspace for 3–5 min. Dissolve the RNA in 15 μL of RNA lysis buffer. The RNA concentration and purity were assessed using a NanoDrop 2000 spectrophotometer. After zeroing the instrument, 2.5 μL of RNA sample was applied to the measurement pedestal, the sample arm was lowered, and absorbance values were recorded via the accompanying software. Samples with concentrations exceeding the detection range were diluted to a final concentration of 200 ng/μL. cDNA synthesis was performed using the G3337 reverse transcription kit. A 20 μL reaction mixture was prepared, gently mixed, and centrifuged before being subjected to the reverse transcription program on a standard PCR instrument. For qPCR, 0.1 mL PCR reaction plates were used, and each reverse transcription product was set up in triplicate according to the standard reaction protocol. After loading, the plates were sealed with PCR sealing film using a plate sealer and centrifuged in a microplate centrifuge. Amplification was conducted on a real-time qPCR instrument. All samples were run with three technical replicates.

ΔΔCT method: A = CT (target gene, test sample) − CT (internal control, test sample);

B = CT (target gene, control sample) − CT (internal control, control sample);

K = A − B;

Fold change expression = 2^−K^.

(Detailed information is provided in Appendix A).

## 3. Results

### 3.1. RNA-Seq Data of Abdominal Adipose Tissue in Kazakh Horses

In this study, 16 transcriptome samples of abdominal adipose tissue from male Kazakh horses were collected. A total of 204,077,436,900 high-quality reads were obtained for LncRNAs, averaging 12,754,839,806 reads per sample, with the Q20 scores exceeding 98%, Q30 scores above 94%, and GC content ranging from 53.42% to 56.34%. For miRNAs, 242,130,213 high-quality reads were generated in total, averaging 15,133,138 reads per sample. After quality filtering, 209,628,058 clean tags remained, averaging 13,101,754 per sample, accounting for 62.34% to 92.56% of clean reads. For circRNAs, 202,566,772,893 clean reads were obtained, averaging 12,660,423,306 reads per sample, with the Q20 scores exceeding 98%, Q30 scores above 94%, and GC content ranging from 53.28% to 56.34% (detailed information is provided in Appendix A).

### 3.2. Correlation Analysis of Adipose Tissue Samples from Male Kazakh Horses

The analysis of gene expression in the adipose tissue samples indicated that, as shown in Figure 1A,C,E, Group W exhibited the highest expression, while Group Y exhibited the lowest. Differences in expression among individual samples were minimal, and the overall expression levels were consistent within groups. Similarly, as depicted in Figure 1B,D,F, the correlation analysis among samples within each group showed similar trends.

### 3.3. Differential Expression Analysis of Adipose Tissue in Male Kazakh Horses

Differential expression analysis of the abdominal adipose tissue in male Kazakh horses yielded the following results. As shown in Figure 2A, a total of 205 DELncRNAs were identified in Group W vs. Group Y including *ENSECAG00000003836*, *ENSECAG00000017858*, and *ENSECAG00000035167*. Among these, 24 DELncRNAs, such as *MSTRG.10991.1* and *MSTRG.7546.5*, were significantly upregulated (*p* < 0.05), while 181 DELncRNAs, such as *ENSECAT00000077193* and *ENSECAT00000050554*, were significantly downregulated (*p* < 0.05). As shown in Figure 2C, a total of 52 DEmiRNAs were detected including *miR-200-y* and *eca-miR-9a*. Of these, 18 DEmiRNAs were significantly upregulated (*p* < 0.05), while 34 DEmiRNAs were significantly downregulated (*p* < 0.05). As shown in Figure 2E, a total of 559 DEcircRNAs were identified between the two groups including *ZNF226* and *ITPKC*. Among them, 237 DEcircRNAs were significantly upregulated (*p* < 0.05), while 322 DEcircRNAs, such as *SAMD4B* and *TIMM50*, were significantly downregulated (*p* < 0.05) (Appendix A). Cluster analysis, as illustrated in Figure 2B,D,F, revealed high reproducibility across biological replicates of the abdominal adipose tissue samples, underscoring the significant transcriptomic differences between Group W and Group Y.

### 3.4. GO Functional Annotation and KEGG Enrichment Analysis of DEGs in Adipose Tissue of Male Kazakh Horses

As shown in Figure 3A, GO annotation of DELncRNAs between Group W and Group Y revealed that these DELncRNAs were primarily associated with the following terms: chemical homeostasis (BP), multicellular organismal water homeostasis (BP), intracellular part (CC), cytoplasmic part (CC), iron-sulfur cluster binding (MF), and metal cluster binding (MF). As shown in Figure 3B, KEGG enrichment analysis of the DELncRNAs between Group W and Group Y indicated that these DELncRNAs were mainly enriched in pathways such as vitamin B6 metabolism, tryptophan metabolism, and glycerolipid metabolism.

As illustrated in Figure 4A, GO annotation of DEmiRNAs in Groups W and Y revealed that the DEmiRNAs were predominantly associated with BP terms such as localization and single-multicellular organism process, CC terms including cell cortex and focal adhesion, and MF terms such as Ras GTPase binding and small GTPase binding. As shown in Figure 4B, KEGG pathway enrichment analysis indicated that these DEmiRNAs were mainly enriched in pathways including fatty acid metabolism, PPAR signaling pathway, and insulin signaling pathway.

As illustrated in Figure 5A, GO annotation of DEcircRNAs between Groups W and Y revealed that these DEcircRNAs were primarily enriched in BP terms such as organelle organization and cellular component organization, CC terms including intracellular and organelle; and MF terms such as catalytic activity and protein binding. As shown in Figure 5B, KEGG pathway enrichment analysis indicated that the DEcircRNAs were predominantly enriched in pathways including the mTOR signaling pathway, AMPK signaling pathway, and the biosynthesis of unsaturated fatty acids (detailed information is provided in Appendix A).

### 3.5. Construction and Analysis of the ceRNA Network

Based on the regulatory interactions between DEcircRNAs–DEmiRNAs and DEmiRNAs–DELncRNAs, co-expression networks involving DELncRNAs and DEcircRNAs mediated by shared DEmiRNAs were identified. As illustrated in Figure 6, within the adipose tissue, a total of 10 circRNA–miRNA–LncRNA interaction axes were discovered, comprising 1 circRNA, 3 miRNAs, and 9 LncRNAs, suggesting that these molecules may act as critical regulators of the color variation in adipose tissue.

### 3.6. PPI Network Analysis

Based on the DEGs in the adipose tissue of male Kazakh horses exhibiting different fat colors, a PPI network was constructed. As illustrated in Figure 7, ten hub genes were identified in both groups including *UBE2L6*, *AKAP4*, *OAZ3*, *TRIM36*, and *PROCA1*.

### 3.7. RT-qPCR Validation

To verify the accuracy of ncRNA sequencing, 6 DELncRNAs and 3 DEmiRNAs were randomly selected for RT-qPCR validation including *ENSECAG00000017858*, *ENSECAG00000035167*, and *ENSECAG00000035873*. As shown in Figure 8, *ENSECAG00000035167*, *ENSECAG00000035873*, and *ENSECAG00000024185* were highly significantly upregulated (*p* < 0.01), while *ENSECAG00000014847* was significantly upregulated (*p* < 0.05). The expression trends of all selected genes were consistent between the RT-qPCR and RNA-Seq results, demonstrating that the sequencing data and expression profiles identified in this study were accurate and reliable, thus suitable for subsequent analyses (see Appendix A).

## 4. Discussion

Adipose tissue exhibits significant functional and structural differences based on its color [22,23] and plays an important role in energy metabolism [24]. White adipocytes primarily store excess calories as triglycerides, which form large lipid droplets occupying the majority of the cell volume [25]. In contrast, brown and beige adipocytes perform functions opposite to those of white adipocytes, mainly dissipating energy through thermogenesis. Compared with white adipocytes, brown and beige adipocytes contain a higher density of mitochondria and exhibit elevated expression of uncoupling protein 1 (*UCP-1*), a key driver of the thermogenic process [26]. Fat color is influenced by multiple factors including breed and age, and varies across anatomical locations [27]. Belhaj et al. [28] conducted an investigation on carcass traits and meat quality involving 208 sheep from the “Beni-Guil” and “Ouled-Djellal” breeds, revealing that in the Ouled-Djellal breed, subcutaneous fat covering the carcass progressively changed color with increasing slaughter age.

### 4.1. Association Between Target Genes of DELncRNAs and Fat Color Regulation

This study showed that target genes of DELncRNAs, including *AOX1* and *LPL*, are predominantly enriched in pathways such as vitamin B6 metabolism and glycerolipid metabolism. As a protein-coding gene belonging to the flavoprotein family, *AOX1* participates in metabolic pathways related to water-soluble vitamins and cofactors. The expression of *AOX1* may regulate the pyridoxal phosphate (PLP) levels by modulating vitamin B6 metabolism, thereby influencing heme biosynthesis and ultimately affecting the color variation of adipose tissue. Vitamin B6, a water-soluble vitamin first discovered during the treatment of dermatitis in rats [29], has PLP as one of its active forms [30]. PLP is involved in over 150 enzymatic reactions [31] and plays an important role in heme biosynthesis [32,33,34,35]. BAT contains numerous mitochondria rich in heme, and heme acts as a prosthetic group for cytochromes, conferring the characteristic brown-red coloration to mitochondria [36]. The high-density accumulation of heme within adipocyte mitochondria accounts for the brown color of the tissue [37]. Furthermore, AOX1 participates in the oxidation of retinal within the metabolism of vitamin A. Retinal, the active form of vitamin A, can be oxidized by AOX1 to retinoic acid [38]. In vivo, all-trans retinoic acid functions as a key endogenous signaling molecule, and AOX1 may play a role in its biosynthesis [39]. Retinoic acid may influence the thermogenic capacity of brown adipocytes by inducing the phosphorylation of p38/MAPK. In vivo, supplementation with all-trans retinoic acid and 9-cis retinoic acid significantly enhances thermogenesis in the brown adipose tissue of mice, and 9-cis retinoic acid can even prevent BAT whitening induced by cold adaptation [40]. Lipoprotein lipase (*LPL*) is abundantly expressed in BAT, and the activation of BAT enhances the hydrolysis of triglycerides in triglyceride-rich lipoproteins (TRLs). When BAT is activated by cold, the expression of angiopoietin-like 4 (*ANGPTL4*), an inhibitor of *LPL* activity and activator of AMPK, is downregulated, thereby increasing plasma *LPL* activity and promoting the uptake of triglyceride-derived fatty acids [41]. WAT primarily stores energy in the form of triglycerides, whereas BAT activation facilitates the efficient metabolism of triglycerides, reducing the WAT content and consequently influencing the color variation of adipose tissue. Bartelt et al. [42] demonstrated that BAT plays an important role in triglyceride clearance, and BAT activation in hypertriglyceridemic mice significantly reduced the plasma levels of triglycerides, providing important insights supporting our findings.

### 4.2. Potential Regulatory Mechanisms Regarding Target Genes of DEmiRNAs in Fat Color Regulation

Extensive research has indicated that differential gene expression affects the color variation in adipose tissue [43,44]. Chou et al. [45] demonstrated that *KSRP* knockout decreases *miR-150* expression, attenuating its suppression of key BAT-related regulators *Prdm16* and *Ppargc1a*. This results in the increased expression of BAT-selective genes in mouse subcutaneous and inguinal WAT, thereby facilitating the browning of WAT. In this study, target genes of DEmiRNAs, including *MAPK1* and *ACACA*, were identified as potential regulators of color variation in adipose tissue. *MAPK1* is implicated in the regulation of adipocyte differentiation and function via the MAPK signaling pathway [46]. This pathway constitutes a complex cascade involving subfamilies such as ERK, JNK, and p38 MAPK [47]. Among these, p38 MAPK plays an important role in adipocyte thermogenesis and enhances mitochondrial function, promoting BAT-associated thermogenic functions and consequently influencing color variation in adipose tissue. In addition, p38 MAPK phosphorylates *PGC-1α* [48], and Kang et al. [49] proposed that *PGC-1α* acts as a key transcriptional coactivator that stimulates mitochondrial DNA replication and gene transcription, thereby driving mitochondrial biogenesis. Furthermore, Bordicchia et al. [50] reported that p38 MAPK enhances the expression of brown adipose and mitochondrial markers. Ventricular infusion of brain natriuretic peptide (BNP) in mice significantly elevates *PGC-1α* and *UCP-1* expression in both WAT and BAT, thereby increasing energy expenditure. *ACACA* is a protein-coding gene belonging to the acetyl-CoA carboxylase family. It plays an important role in pathways related to water-soluble vitamin and cofactor metabolism, steroid metabolism, and is one of the key genes involved in fatty acid metabolism. [51] Lee et al. [52] found that sesamol suppresses the expression of genes related to white adipogenesis including *ACACA*, *PPARγ*, and *SREBP-1c*, reduces lipid droplet content in BAT, and upregulates BAT marker genes *UCP-1*, *FGF21*, and *COXII*, thereby promoting the browning of WAT.

### 4.3. Association Between Target Genes of DEcircRNAs and Pathways Related to Color Variation in Adipose Tissue

This study revealed that multiple target genes of DEcircRNA are significantly enriched in the mTOR signaling pathway. This pathway is a key regulator of biological processes such as adipogenesis, lipid metabolism, and thermogenesis [53]. Both mTOR complexes, mTORC1 and mTORC2, play important roles in adipocyte differentiation [54]. Considering that BAT exhibits a darker color compared with WAT, the mTOR signaling pathway may influence adipose browning, leading to a darkening of the WAT color. Zhang et al. [55] confirmed that mTORC1 acts as an effective regulator of WAT browning by phosphorylating *CREB*-regulated transcription coactivator 2 (*CRTC2*), which suppresses cyclooxygenase-2 (COX-2), thereby inhibiting the prostaglandin signaling pathway associated with WAT browning. Consequently, the inhibition of mTORC1 results in the darkening of WAT. Liu et al. [56] demonstrated that cold exposure (4 °C) in mice induced the activation of mTORC1 in BAT and WAT. Rapamycin treatment inhibits this cold-induced activation and regulates *UCP-1* expression, thereby reducing the thermogenic capacity of BAT and hindering WAT browning. Moreover, treatment with the β3-adrenergic receptor agonist CL316,243 induces the formation of multilocular adipocytes in WAT, enhances *UCP-1* expression, promotes mitochondrial biogenesis, and elicits other phenotypic features related to color variation in adipose tissue.

### 4.4. Potential Regulatory Mechanisms of the ceRNA Network

Based on the unique regulatory relationships among DELncRNAs, DEmiRNAs, and DEmRNAs, this study constructed a ceRNA network in the abdominal adipose tissue of Kazakh horses. The results suggest that the novel_circ_023251–eca-miR-902–ENSECAG00000017793 axis and the novel_circ_023251–miR-7-x–ENSECAG00000007276 axis may hold potential in regulating the color of adipose tissue. *NGFR* is a transmembrane protein that primarily mediates nerve growth factor (NGF) signaling [57], and novel_circ_023251–eca-miR-902–ENSECAG00000017793 may functionally interact with *NGFR* to influence the color variation in adipose tissue. The PI3K/Akt pathway is a principal downstream signaling cascade of NGF [58], and the PI3K/Akt/mTOR signaling pathway is critical for regulating cell survival, proliferation, and metabolism [59]. Wang et al. [60] demonstrated that deletion of the RIIβ subunit of protein kinase A (PKA) enhanced sympathetic nerve activity in the WAT of mice and activated mTOR signaling. This activation promotes the expression of browning-related genes such as *UCP-1* and *PGC-1α* and increases mitochondrial activity and energy expenditure, ultimately inducing WAT browning. The inhibition of mTOR abolishes the browning phenotype in RIIβ-deficient mice. The regulation of *EPHB6* expression may be modulated by the interaction between novel_circ_023251, miR-7-x, and ENSECAG00000007276, suggesting a potential functional association. The novel_circ_023251–miR-7-x–ENSECAG00000007276 axis may regulate *EPHB6* expression, thereby modulating vitamin B6 metabolism, maintaining mitochondrial function, and influencing the color of adipose tissue. Vitamin B6 is essential for normal mitochondrial and cellular function [61]. Li et al. [62] found that *EPHB6* deficiency in mice causes gut microbiota dysbiosis and disrupts the homeostasis of vitamin B6 metabolism, characterized by increased pyridoxine (PN) in feces and decreased pyridoxamine (PM) and PLP in plasma, alongside reduced PLP in the prefrontal cortex (PFC). Transplantation of fecal microbiota from wild-type mice into *EPHB6*-deficient mice normalized PN in feces and increased PM and PLP in plasma as well as PLP in the PFC, indicating that *EPHB6* influences the vitamin B6 levels in feces, blood, and PFC through regulation of the gut microbiota.

## 5. Conclusions

In this study, transcriptomic sequencing analysis was conducted on different colors of adipose tissue in male Kazakh horses. The results revealed that DEGs, such as *AOX1*, *LPL*, *MAPK1*, and *ACACA*, were enriched in pathways including vitamin B6 metabolism, glycerolipid metabolism, and the MAPK and mTOR signaling pathways. Moreover, these DEGs interacted with regulatory axes such as novel_circ_023251–eca-miR-902–ENSECAG00000017793 and novel_circ_023251–miR-7-x–ENSECAG00000007276, playing important roles in pathways related to lipid synthesis and metabolism. These interactions collectively regulate the development and color variation of adipose tissue in Kazakh horses. Our research team is currently performing protein expression analyses and in vitro validation experiments for the genes *AOX1*, *LPL*, *MAPK1*, and *ACACA* to further clarify their functional roles.

## Figures and Tables

**Figure 1 biology-14-01285-f001:**
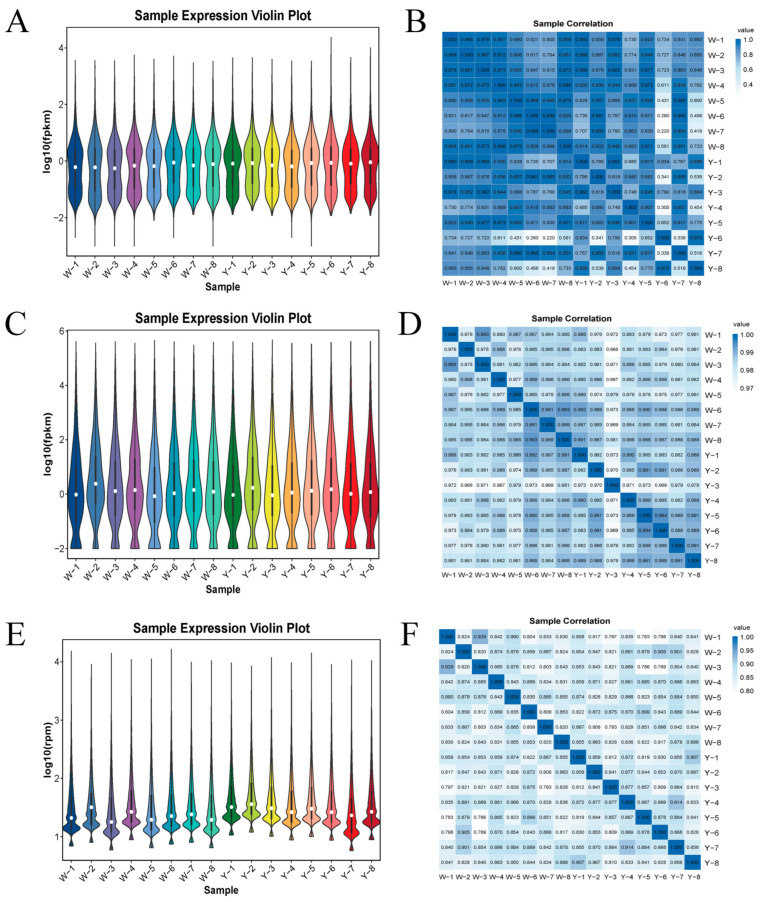
Violin plots of the expression and correlation heatmaps of samples in Group W vs. Group Y. (**A**) Violin plot of LncRNA expression in Group W vs. Group Y; (**B**) correlation heatmap of LncRNAs in Group W vs. Group Y; (**C**) violin plot of miRNA expression in Group W vs. Group Y; (**D**) correlation heatmap of miRNAs in Group W vs. Group Y; (**E**) violin plot of circRNA expression in Group W vs. Group Y; (**F**) correlation heatmap of circRNAs in Group W vs. Group Y.

**Figure 2 biology-14-01285-f002:**
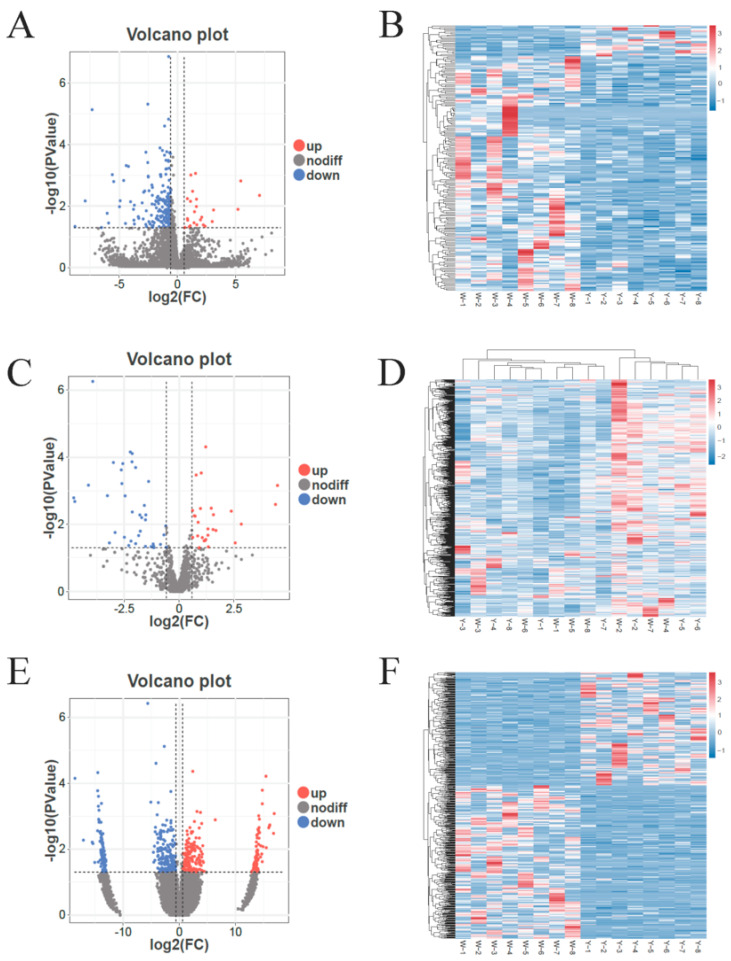
Volcano plots and hierarchical clustering analyses in Group W vs. Group Y. (**A**) Volcano plot of DELncRNAs in Group W vs. Group Y; (**B**) hierarchical clustering of DELncRNAs in Group W vs. Group Y; (**C**) volcano plot of DEmiRNAs in Group W vs. Group Y; (**D**) hierarchical clustering of DEmiRNAs in Group W vs. Group Y; (**E**) volcano plot of DEcirRNAs in Group W vs. Group Y; (**F**) hierarchical clustering of DEcirRNAs in Group W vs. Group Y. Note: In Figure (**A**,**C**,**E**), “up” and “down” indicate upregulated and downregulated gene expression, respectively, in the adipose tissue of male Kazakh horses. In Figure (**B**,**D**,**F**), the horizontal axis represents individual samples, the vertical axis represents the levels of expression, and the color gradient from blue to red indicates a gradual increase in expression.

**Figure 3 biology-14-01285-f003:**
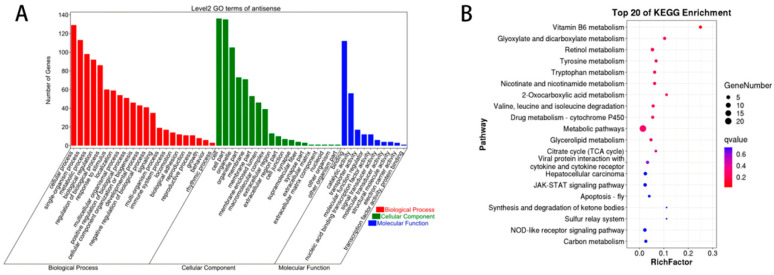
GO annotation and KEGG enrichment analysis of DELncRNAs. (**A**) GO annotation terms of DELncRNAs; (**B**) KEGG pathway enrichment analysis of DELncRNAs.

**Figure 4 biology-14-01285-f004:**
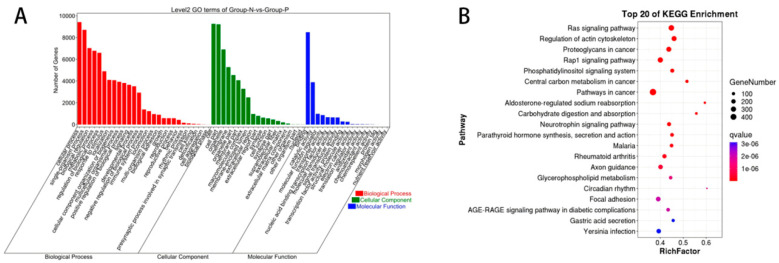
GO annotation and KEGG enrichment analysis of DEmiRNAs. (**A**) GO annotation terms of DEmiRNAs; (**B**) KEGG pathway enrichment analysis of DEmiRNAs.

**Figure 5 biology-14-01285-f005:**
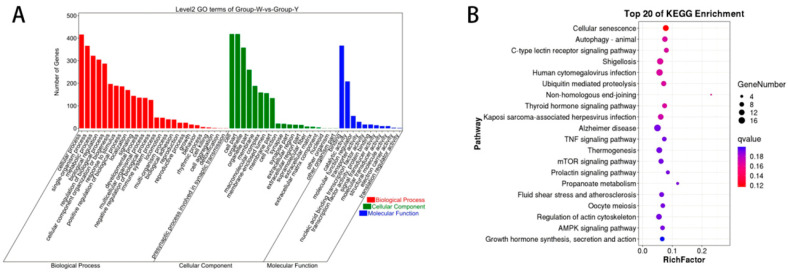
GO annotation and KEGG enrichment analysis of DEcircRNAs. (**A**) GO annotation terms of DEcircRNAs; (**B**) KEGG pathway enrichment analysis of DEcircRNAs. Note: In Figure 3A, Figure 4A and Figure 5A, the x-axis represents individual DEGs. Red bars represent BP, green bars represent CC, and blue bars represent MF. Figure 3B, Figure 4B and Figure 5B indicate the top 20 pathways ranked by the lowest Q-values. The y-axis indicates the pathway names, and the x-axis indicates the proportion of the gene. “Count” represents the number of genes associated with each pathway, while the color gradient from blue to red reflects decreasing Q-values.

**Figure 6 biology-14-01285-f006:**
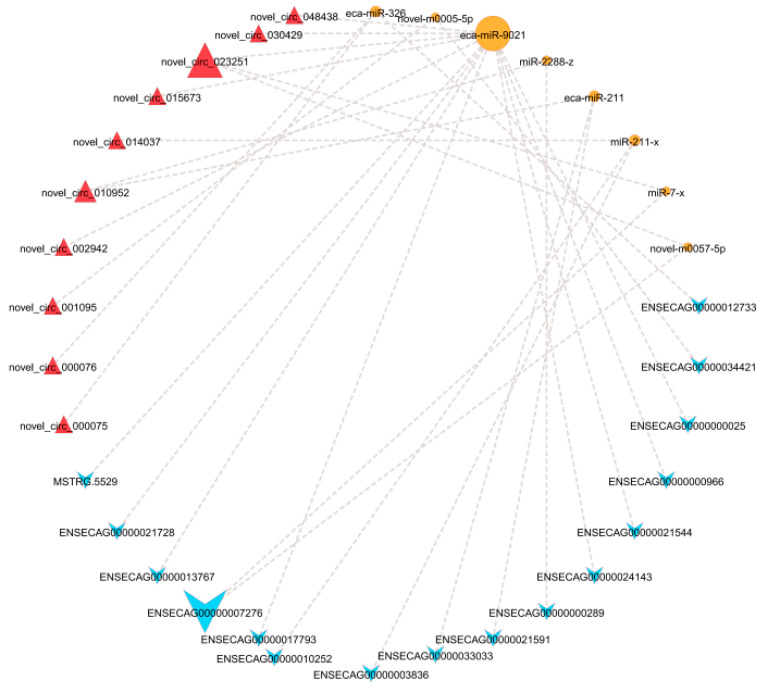
The circRNA–miRNA–LncRNA interaction network. Note: Red triangles represent circRNAs, orange circles represent miRNAs, and blue triangles represent LncRNAs.

**Figure 7 biology-14-01285-f007:**
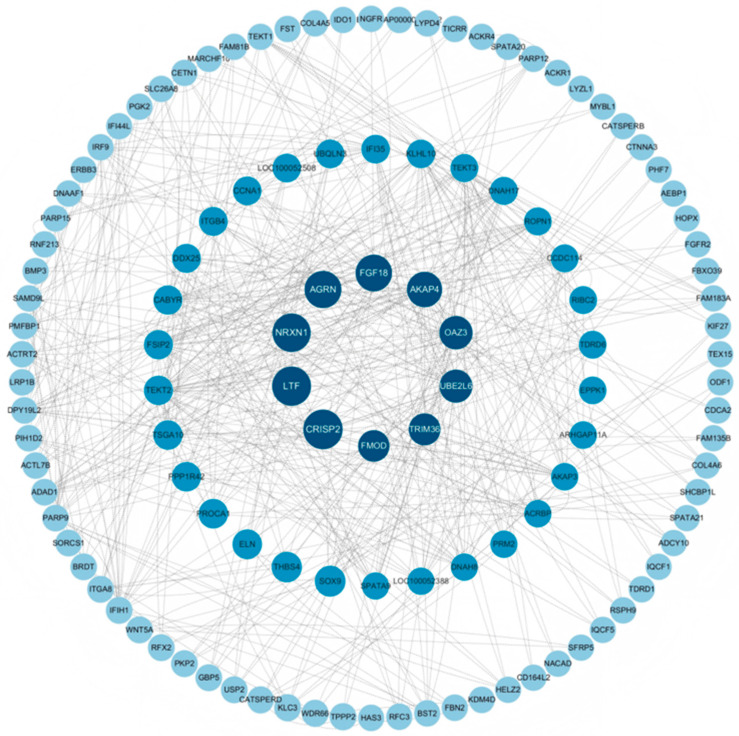
Visualization of PPI network.

**Figure 8 biology-14-01285-f008:**
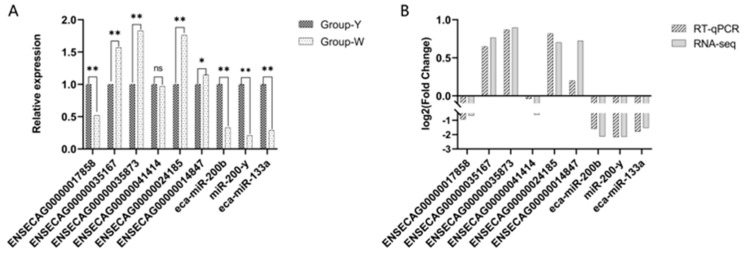
RT-qPCR validation. (**A**) Differential gene expression in Group Y vs. Group W detected by RT-qPCR; (**B**) comparison of Log2 fold change (Log2FC) of DEGs in Group Y vs. Group W as measured by RNA-Seq and RT-qPCR. Note: In the figure, “*” indicates a significant difference (*p* < 0.05), and “**” indicates a highly significant difference (*p* < 0.01).

## Data Availability

The data presented in this study are openly available in BioProject with reference number PRJNA1295436.

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
