# Peer review of "Mechanisms of Variation in Abdominal Adipose Color Among Male Kazakh Horses Through Non-Coding RNA Sequencing"

_biology, 2025, doi:10.3390/biology14091285_

Round 1
Reviewer 1 Report
Comments and Suggestions for Authors
Firstly, this manuscript is overall very good and highly valuable for research. The author takes the mechanism of fat color production as the starting point and conducts in-depth exploration through transcriptome sequencing technology and bioinformatics analysis. However, I still have the following questions that need to be answered by the author: However, I still have the following questions that need to be answered by the author:
- The Materials and Methods chapter lacks a detailed description of the tools and parameters used in the analysis process.
- How was the feeding situation of the experimental animals before slaughter? Please provide the feeding information.
- The Materials and Methods section lacks detailed information on library preparation.
- The author should add detailed information about the PCR method in the Materials and Methods (cDNA synthesis, primer selection, PCR, sample quantity, repetition times, statistics).
- In the manuscript, if P < 0.05, then P should be italicized, for example, in lines 333 and 335, etc.
- Please check if the genes in the entire text are italicized.
Author Response
Response to Reviewer 1
Comments 1: The Materials and Methods chapter lacks a detailed description of the tools and parameters used in the analysis process.
Response: Thank you very much for your valuable suggestions. Your expert guidance has played a critical role in helping us improve the quality of our manuscript.We have supplemented the detailed descriptions of the tools and parameters used in the analysis process, such as “2.3 Quality control and assessment of data” and “2.4 Differential expression analysis”. As both reviewers raised this issue, the revised content has been highlighted in yellow and blue in the original text.
The revised version is now expressed as follows:
“2.3 Quality control and assessment of data” (line 127-136, page 4)
Raw sequencing data in FASTQ format, containing adapter sequences and low-quality reads, were obtained directly from the sequencer. To obtain high-quality clean reads, filtering was performed using fastp (v 0.23.4) with the following parameters: (1) removal of reads containing adapter sequences; (2) removal of reads with >10% ambiguous bases (N); and (3) removal of low-quality reads in which bases with a Phred quality score ≤20 accounted for >50% of the sequence [21]. Differential expression analysis of lncRNA, miRNA, and circRNA transcripts was conducted, and significantly differentially expressed transcripts were identified using the criteria |log2(fold change)| ≥ 1.5 and P < 0.05, with multiple testing correction applied to P-values using q < 1.00.
“2.4 Differential expression analysis” (line 137-142, page 4)
For each sample, aligned reads were assembled using a reference-based method with StringTie v2.2.1. For each transcript region, FPKM (fragments per kilobase of transcript per million mapped reads) values were calculated to quantify gene expression levels and their variability. RNAs with P < 0.05 and an absolute fold change greater than 1 were considered significantly differentially expressed.
Comments 2:How was the feeding situation of the experimental animals before slaughter? Please provide the feeding information.
Response: Thank you very much for your valuable suggestions. Your expert guidance has played a critical role in helping us improve the quality of our manuscript. We have supplemented the feeding conditions of the experimental animals before slaughter in the Sample collection. The modified content has been highlighted in yellow in the original text.
The revised version is now expressed as follows:
“2.1 Sample collection” (line 93-104, page 3)
In 2025, 16 adult male Kazakh horses aged 6 to 8 years were randomly selected from the Wantong Animal Husbandry Co., Ltd., Emin County. Abdominal adipose tissue samples were collected from these animals. According to the color, the samples were categorized into two groups, i.e., Group W (white adipose tissue) and Group Y (yellow adipose tissue), with eight biological replicates in each group. All experimental horses were fed high-quality alfalfa hay and corn kernels, with ad libitum access to water. All horses were raised, managed, fattened, and slaughtered under uniform standardized conditions. Immediately post-slaughter, abdominal adipose tissue samples measuring 1 cm × 1 cm × 1 cm were excised from the left half of the carcass, placed into cryogenic vials with two aliquots per sample, rapidly frozen in liquid nitrogen, and stored at –80 °C for analyses.
Comments 3:The Materials and Methods section lacks detailed information on library preparation.
Response: Thank you very much for your valuable suggestions. Your expert guidance has played a critical role in helping us improve the quality of our manuscript. We have supplemented the detailed information on library preparation in the materials and methods. As both reviewers raised this issue, the revised content has been highlighted in yellow and blue in the original text.
The revised version is now expressed as follows:
“2.2 Non-coding RNA sequencing” (line 108-126, page 3)
Total RNA was extracted from each adipose tissue sample using TRIzol reagent (Thermo Fisher Scientific, Carlsbad, CA, USA). RNA concentration and integrity were evaluated with a NanoDrop microvolume spectrophotometer (Thermo Fisher Scientific, Carlsbad, CA, USA), an Agilent 2100 Bioanalyzer (Agilent Technologies, Santa Clara, CA, USA), and 1% agarose gel electrophoresis. Ribosomal RNA (rRNA) was removed using the Ribo-off rRNA Depletion Kit (Vazyme, Nanjing, Jiangsu, China), and sequencing libraries were prepared with the VAHTS Universal V6 RNA-seq Library Prep Kit (Vazyme, Nanjing, Jiangsu, China). The main steps were as follows: after hybridization of RNA with probes, rRNA was digested by RNase H and DNase I, and the resulting rRNA-depleted RNA was purified with VAHTS RNA Clean Beads; the purified RNA was then fragmented and used for first-strand cDNA synthesis with random primers, followed by second-strand cDNA synthesis in a dUTP-containing system; Illumina adapters were ligated using Rapid DNA Ligase, and the ligated products were purified twice with VAHTS DNA Clean Beads; PCR amplification was carried out with a premix containing UDG enzyme to enrich adapter-ligated fragments while degrading the second-strand cDNA; the amplified products were further purified with magnetic beads to generate high-quality libraries. Library quality was assessed using the High Sensitivity DNA Assay Kit (Agilent Technologies, Santa Clara, CA, USA), and precise quantification was performed via qPCR. Sequencing was conducted by Kidio Biotechnology Co., Ltd. (Guangzhou, China)[20].
Comments 4:The author should add detailed information about the PCR method in the Materials and Methods (cDNA synthesis, primer selection, PCR, sample quantity, repetition times, statistics).
Response: Thank you very much for your valuable suggestions. Your expert guidance has played a critical role in helping us improve the quality of our manuscript. We have added detailed information about the PCR method in the Materials and Methods section. The modified content has been highlighted in yellow in the original text.
The revised version is now expressed as follows:
“2.7 RT-qPCR validation”(line 158- 189, page 5)
To validate selected LncRNAs, miRNAs, and circRNAs, total RNA was reverse-transcribed into complementary DNA (cDNA). The procedure was conducted as follows: Pre-chill a grinding tube on ice and add 1 mL of RNA extraction reagent along with three 3 mm grinding beads. Add 5–20 mg of tissue and thoroughly homogenize until no visible tissue fragments remain. Centrifuge the homogenate at 12,000 rpm and 4°C for 10 min and collect the supernatant. Add 100 μL of chloroform substitute, invert the tube for 15 s to mix, and incubate at room temperature for 3 min. Centrifuge again at 12,000 rpm and 4°C for 10 min, transfer 400 μL of the upper aqueous phase to a new tube, and add 550 μL of isopropanol, gently inverting to mix. Incubate at –20°C for 15 min. Centrifuge at 12,000 rpm and 4°C for 10 min; a white pellet at the bottom represents RNA. Discard the supernatant and wash the pellet with 1 mL of 75% ethanol, centrifuge at 12,000 rpm and 4°C for 5 min, and repeat the wash once. Completely remove the ethanol and air-dry the pellet in a sterile workspace for 3–5 min. Dissolve the RNA in 15 μL of RNA lysis buffer. RNA concentration and purity were assessed using a NanoDrop 2000 spectrophotometer. After zeroing the instrument, 2.5 μL of RNA sample was applied to the measurement pedestal, the sample arm was lowered, and absorbance values were recorded via the accompanying software. Samples with concentrations exceeding the detection range were diluted to a final concentration of 200 ng/μL. cDNA synthesis was performed using the G3337 reverse transcription kit. A 20 μL reaction mixture was prepared, gently mixed, and centrifuged before being subjected to the reverse transcription program on a standard PCR instrument. For qPCR, 0.1 mL PCR reaction plates were used, and each reverse transcription product was set up in triplicate according to the standard reaction protocol. After loading, the plates were sealed with PCR sealing film using a plate sealer and centrifuged in a microplate centrifuge. Amplification was conducted on a real-time qPCR instrument. All samples were run with three technical replicates.
ΔΔCT method: A = CT (target gene, test sample) - CT (internal control, test sample)
B = CT (target gene, control sample) - CT (internal control, control sample)
(Detailed information is provided in Supplementary Material 1.)
Comments 5:In the manuscript, if P < 0.05, then P should be italicized, for example, in lines 333 and 335, etc.
Response: Thank you very much for your valuable suggestions. Your expert guidance has played a critical role in helping us improve the quality of our manuscript. We have italicized the “P” in the entire text. The modified content has been highlighted in yellow in the original text.
Comments 6:Please check if the genes in the entire text are italicized.
Response: Thank you very much for your valuable suggestions. Your expert guidance has played a critical role in helping us improve the quality of our manuscript. We have checked and italicized the genes in the full text.As both reviewers raised this issue, the revised content has been highlighted in yellow and blue in the original text.

Reviewer 2 Report
Comments and Suggestions for Authors
Currently, the share of horse meat production in the market is not high, which leads to a scarcity of research on horse meat and even fewer studies on horse fat. The author is able to conduct in-depth exploration based on the industry and use molecular biology to explain complex biological processes. The overall quality of the research is good in terms of significance, purpose, and the scientific questions to be addressed. However, I suggest the author make the following revisions:
- Please provide more detailed information in the Materials and Methods section.
- Please add a description of the non-coding RNA data analysis workflow in the text. The sequencing parameters (such as sequencing platform model, read length, sequencing depth, etc.) have not been mentioned.
- What does the * symbol in Figure 9 represent? Please add the annotation.
- Please check and ensure that all gene names are in italics throughout the manuscript.
- Conduct functional analysis of the genes to confirm their roles.
- The conclusion should mention that further experiments should be conducted to measure the protein levels of these genes and their functional roles.
Author Response
Response to Reviewer 2
Comments 1: The Materials and Methods chapter lacks a detailed description of the tools and parameters used in the analysis process.
Response: Thank you very much for your valuable suggestions. Your expert guidance has played a critical role in helping us improve the quality of our manuscript. We have revised this part. Such as “2.1 Sample collection” and “2.2 Non-coding RNA sequencing”. As both reviewers raised this issue, the revised content has been highlighted in yellow and blue in the original text.
The revised version is now expressed as follows:
“2.1 Sample collection” (line94-104, page 3)
In 2025, 16 adult male Kazakh horses aged 6 to 8 years were randomly selected from the Wantong Animal Husbandry Co., Ltd., Emin County. Abdominal adipose tissue samples were collected from these animals. According to the color, the samples were categorized into two groups, i.e., Group W (white adipose tissue) and Group Y (yellow adipose tissue), with eight biological replicates in each group. All experimental horses were fed high-quality alfalfa hay and corn kernels, with ad libitum access to water. All horses were raised, managed, fattened, and slaughtered under uniform standardized conditions. Immediately post-slaughter, abdominal adipose tissue samples measuring 1 cm × 1 cm × 1 cm were excised from the left half of the carcass, placed into cryogenic vials with two aliquots per sample, rapidly frozen in liquid nitrogen, and stored at –80 °C for analyses.
“2.2 Non-coding RNA sequencing” (line 108-126, page 3)
Total RNA was extracted from each adipose tissue sample using TRIzol reagent (Thermo Fisher Scientific, Carlsbad, CA, USA). RNA concentration and integrity were evaluated with a NanoDrop microvolume spectrophotometer (Thermo Fisher Scientific, Carlsbad, CA, USA), an Agilent 2100 Bioanalyzer (Agilent Technologies, Santa Clara, CA, USA), and 1% agarose gel electrophoresis. Ribosomal RNA (rRNA) was removed using the Ribo-off rRNA Depletion Kit (Vazyme, Nanjing, Jiangsu, China), and sequencing libraries were prepared with the VAHTS Universal V6 RNA-seq Library Prep Kit (Vazyme, Nanjing, Jiangsu, China). The main steps were as follows: after hybridization of RNA with probes, rRNA was digested by RNase H and DNase I, and the resulting rRNA-depleted RNA was purified with VAHTS RNA Clean Beads; the purified RNA was then fragmented and used for first-strand cDNA synthesis with random primers, followed by second-strand cDNA synthesis in a dUTP-containing system; Illumina adapters were ligated using Rapid DNA Ligase, and the ligated products were purified twice with VAHTS DNA Clean Beads; PCR amplification was carried out with a premix containing UDG enzyme to enrich adapter-ligated fragments while degrading the second-strand cDNA; the amplified products were further purified with magnetic beads to generate high-quality libraries. Library quality was assessed using the High Sensitivity DNA Assay Kit (Agilent Technologies, Santa Clara, CA, USA), and precise quantification was performed via qPCR. Sequencing was conducted by Kidio Biotechnology Co., Ltd. (Guangzhou, China)[20].
Comments 2: Please add a description of the non-coding RNA data analysis workflow in the text. The sequencing parameters (such as sequencing platform model, read length, sequencing depth, etc.) have not been mentioned.
Response: Thank you very much for your valuable suggestions. Your expert guidance has played a critical role in helping us improve the quality of our manuscript. We have added a description of the non-coding RNA data analysis workflow in the text. As both reviewers raised this issue, the revised content has been highlighted in yellow and blue in the original text.
The revised version is now expressed as follows:
“2.3 Quality control and assessment of data” (line 127- 136, page 4)
Raw sequencing data in FASTQ format, containing adapter sequences and low-quality reads, were obtained directly from the sequencer. To obtain high-quality clean reads, filtering was performed using fastp (v 0.23.4) with the following parameters: (1) removal of reads containing adapter sequences; (2) removal of reads with >10% ambiguous bases (N); and (3) removal of low-quality reads in which bases with a Phred quality score ≤20 accounted for >50% of the sequence [21]. Differential expression analysis of lncRNA, miRNA, and circRNA transcripts was conducted, and significantly differentially expressed transcripts were identified using the criteria |log2(fold change)| ≥ 1.5 and P < 0.05, with multiple testing correction applied to P-values using q < 1.00.
“2.4 Differential expression analysis” (line 137- 142, page 4)
For each sample, aligned reads were assembled using a reference-based method with StringTie v2.2.1. For each transcript region, FPKM (fragments per kilobase of transcript per million mapped reads) values were calculated to quantify gene expression levels and their variability. RNAs with P < 0.05 and an absolute fold change greater than 1 were considered significantly differentially expressed.
Comments 3: What does the * symbol in Figure 9 represent? Please add the annotation.
Response: Thank you very much for your valuable suggestions. Your expert guidance has played a critical role in helping us improve the quality of our manuscript. We have supplemented the meaning represented by the * in Figure 9. The modified content has been highlighted in blue in the original text.
The revised version is now expressed as follows: (line 323- 324, page 13)
Note: In the figure, “*” indicates a significant difference (P < 0.05), and “**” indicates a highly significant difference (P < 0.01).
Comments 4: Please check and ensure that all gene names are in italics throughout the manuscript.
Response: Thank you very much for your valuable suggestions. Your expert guidance has played a critical role in helping us improve the quality of our manuscript. We have checked and italicized the genes in the full text. As both reviewers raised this issue, the revised content has been highlighted in yellow and blue in the original text.
Comments 5: Conduct functional analysis of the genes to confirm their roles.
Response: Thank you very much for your valuable suggestions. Your expert guidance has played a critical role in helping us improve the quality of our manuscript. We have supplemented the analysis of genes in the discussion section to confirm their functions. The modified content has been highlighted in blue in the original text.
The revised version is now expressed as follows:
4.1 Association between target genes of DELncRNAs and fat color regulation (line 339- 371, page 13)
This study shows that target genes of DELncRNAs, including AOX1 and LPL, are predominantly enriched in pathways such as Vitamin B6 metabolism and Glycerolipid metabolism. As a protein-coding gene belonging to the flavoprotein family, AOX1 participates in metabolic pathways related to water-soluble vitamins and cofactors. The expression of AOX1 may regulate pyridoxal phosphate (PLP) levels by modulating Vitamin B6 metabolism, thereby influencing heme biosynthesis and ultimately affecting the color variation of adipose tissue. Vitamin B6, a water-soluble vitamin first discovered during the treatment of dermatitis in rats[29], has PLP as one of its active forms[30]. PLP is involved in over 150 enzymatic reactions[31] and plays an important role in heme biosynthesis[32-35]. BAT contains numerous mitochondria rich in heme, and heme acts as a prosthetic group for cytochromes, conferring the characteristic brown-red coloration to mitochondria[36]. The high-density accumulation of heme within adipocyte mitochondria accounts for the brown color of the tissue[37]. Furthermore, AOX1 participates in the oxidation of retinal within the metabolism of vitamin A. Retinal, the active form of vitamin A, can be oxidized by AOX1 to retinoic acid [38]. In vivo, all-trans retinoic acid functions as a key endogenous signaling molecule, and AOX1 may play a role in its biosynthesis[39]. Retinoic acid may influence the thermogenic capacity of brown adipocytes by inducing phosphorylation of p38/MAPK. In vivo, supplementation with all-trans retinoic acid and 9-cis retinoic acid significantly enhances thermogenesis in brown adipose tissue of mice, and 9-cis retinoic acid can even prevent BAT whitening induced by cold adaptation[40]. Lipoprotein lipase (LPL) is abundantly expressed in BAT, and the activation of BAT enhances the hydrolysis of triglycerides in triglyceride-rich lipoproteins (TRL). When BAT is activated by cold, the expression of angiopoietin-like 4 (ANGPTL4), an inhibitor of LPL activity and activator of AMPK, is downregulated, thereby increasing plasma LPL activity and promoting uptake of triglyceride-derived fatty acids[41]. WAT primarily stores energy in the form of triglycerides, whereas BAT activation facilitates efficient metabolism of triglycerides, reducing WAT content and consequently influencing the color variation of adipose tissue. Bartelt et al.[42] demonstrated that BAT plays an important role in triglyceride clearance, and BAT activation in hypertriglyceridemic mice significantly reduced plasma levels of triglycerides, providing important insights supporting our findings.
Comments 6: The conclusion should mention that further experiments should be conducted to measure the protein levels of these genes and their functional roles.
Response: Thank you very much for your valuable suggestions. Your expert guidance has played a critical role in helping us improve the quality of our manuscript. We have supplemented the limitations of this study in the conclusion section. The modified content has been highlighted in blue in the original text.
The revised version is now expressed as follows:
“5. Conclusions” (line 450- 461, page 15)
In this study, transcriptomic sequencing analysis was conducted on different colors of adipose tissue in male Kazakh horses. The results reveal that DEGs, such as AOX1, LPL, MAPK1, and ACACA, are enriched in pathways including vitamin B6 metabolism, glycerolipid metabolism, and the MAPK and mTOR signaling pathways. Moreover, these DEGs interact with regulatory axes such as novel_circ_023251–eca-miR-902–ENSECAG00000017793 and novel_circ_023251–miR-7-x–ENSECAG00000007276, playing important roles in pathways related to lipid synthesis and metabolism. These interactions collectively regulate the development and color variation of adipose tissue in Kazakh horses. Our research team is currently performing protein expression analyses and in vitro validation experiments for the genes AOX1, LPL, MAPK1, and ACACA to further clarify their functional roles.

Round 2
Reviewer 2 Report
Comments and Suggestions for Authors
The comments have been addressed.